



# African Anthropogenic Emissions Inventory for gases and particles from 1990 to 2015

Sekou Keita[1], Catherine Liousse[2], Eric-Michel Assamoi[3], Thierno Doumbia[2], N'Datchoh Evelyne Touré[3], Sylvain Gnamien[3], Nellie Elguindi[2], Claire Granier[2,4], Véronique Yoboué[3]

[1]Université Péléforo Gon Coulibaly de Korhogo, UFR Sciences Biologiques, Département Math-Physique-Chimie, BP 1328 Korhogo, Côte d'Ivoire
[2]Laboratoire d'Aérologie, Université Paul Sabatier Toulouse III CNRS, France
[3]Université Félix Houphouët-Boigny, LAPA-MF, BPV34, Abidjan 01, Côte D'Ivoire
[4]NOAA Chemical Sciences Laboratory- CIRES/University of Colorado, Boulder, CO, USA

*Correspondence to*: Sekou Keita (sekkeith@yahoo.fr)

**Abstract.** There are very few African regional inventories providing biofuel and fossil fuel emissions. Within the framework of the DACCIWA project, we have developed an African regional anthropogenic emission inventory including the main
African polluting sources (wood and charcoal burning, charcoal making, truck, car, buses and two wheels vehicles, open waste burning and flaring). To this end, a database on fuel consumption and emission factors specific to Africa was established, using the most recent measurements. New spatial proxies (road network, power plant geographical coordinates) were used to convert national emissions into gridded inventories at a 0.1° x 0.1° spatial resolution. This inventory includes carbonaceous particles (black and organic carbon) and gaseous species (CO, NOx, $SO_2$ and NMVOC) for the period 1990-2015 with a yearly temporal
resolution. We show that all pollutant emissions are globally increasing in Africa during the period 1990-2015 with a growth rate of 95%, 86%, 113%, 112%, 97%, and 130% for BC, OC, NOx, CO, $SO_2$ and NMVOC, respectively. We also show that West Africa is the highest emitting region of BC, OC, CO and NMVOC, followed by East Africa, largely due to domestic fire and traffic activities, while Southern Africa and Northern Africa are the highest emitting regions of $SO_2$ and NOx due to industrial and power plant sources. Emissions from this inventory are compared to other regional and global inventories and
its uncertainties are quantified by a Monte Carlo simulation. Finally, this inventory highlights key pollutant emission sectors in which mitigation scenarios should focus on. The DACCIWA inventory (https://doi.org/10.25326/56, Keita et al., 2017) including the annual gridded emission inventory for Africa for the period 1990-2015 are distributed from the Emissions of atmospheric Compounds and Compilation of Ancillary Data (ECCAD) system (https://eccad.aeris-data.fr/). For review purposes, ECCAD has set up an anonymous repository where subsets of the DACCIWA data can be accessed directly
https://www7.obs-mip.fr/eccad/essd-surf-emis-dacciwa/.



## 1 Introduction

According to the UN (2015) report, "World Population Prospects: The 2015 Revision", Africa is expected to account for more than half of the world's population growth between 2015 and 2050,. This rapid increase in population is accompanied by a dramatic increase in anthropogenic emissions of atmospheric pollutants as shown in Liousse et al., (2014).

Pollutant concentration measurements carried out during POLCA (Liousse and Galy-Lacaux, 2010) show that the air in African urban areas such as Bamako (Mali) and Dakar (Senegal) is already highly polluted and affecting the population's health (Doumbia et al., 2012; Val et al., 2013), and therefore the economy of the region. PM2.5 measurements recently performed as part of the DACCIWA program for Cotonou (Benin) and Abidjan (Cote d'Ivoire) also show that concentrations are 2 to 10 times higher than the WHO standards (Adon et al., 2020; Djossou et al., 2018; Evans et al., 2018). The same results were also

observed in Dakar (Dieme et al., 2012). If nothing is done, air pollution in Africa will worsen because emission regulations are still weak on the continent (Liousse et al., 2014).

Anthropogenic emission inventories are fundamental, not only in order to accurately model air quality and climate change, but also for the development of control and mitigation strategies. Emission inventories are commonly constructed using a bottom-up approach where available statistics on fuel combustion for anthropogenic sources (e.g. traffic, industry, residential

combustion, etc.) are combined with the most representative emission factors. Many of the inventories that exist today are at the global scale and do not contain detailed informations specific to Africa. Nevertheless, such global inventories (Bond et al., 2004; Junker and Liousse, 2008; Granier et al., 2011; Smith et al., 2011; Klimont et al., 2013; Klimont et al., 2017; Hoesly et al., 2018) are used for air quality and climate modelling in Africa (Deroubaix et al., 2018; Haslett et al., 2019).

The few regional inventories that have been published for Africa such as Liousse et al. (2014) for combustion sources and

Assamoi and Liousse (2010) for two-wheeled vehicles, have shown that much uncertainty still remains in our knowledge of fuel consumption, emission factors, spatial distribution of emissions in Africa. Furthermore, they indicate that some important sources are not represented at all. It is therefore a challenge for policy makers to identify specific emission sources in Africa which is necessary to target for designing effective pollution control regulations and mitigation strategies.

This paper presents a comprehensive, consistent and spatially distributed new inventory for the entire of Africa, focused on

the emissions of particles, i.e. black carbon (BC) and organic carbon (OC), and gaseous compounds, namely carbon monoxide (CO), nitrogen oxides (NOx), sulfur dioxide ($SO_2$) and non-methane volatile organic compounds (NMVOC). To our knowledge, this inventory covers the longest period (1990-2015) and considers the main anthropogenic emissions sources specific to Africa, such as open waste burning, charcoal making, flaring emissions as described in Doumbia et al. (2019) and two-wheeled vehicles emissions as described in Assamoi and Liousse (2010), in addition to traffic, domestics fires, industries

and power plants. It takes into account new emission factors reported by Keita et al. (2018).

Section 2 describes the methodology and data sources selected for the different emission sources. The results for sectoral emissions, spatial distributions, as well as emission trends are presented in Section 3, which also includes a comparison with other studies together with a discussion on uncertainties.



## 2 Methodology

The quantification of biofuel and fossil fuel emission inventories from 1990 to 2015 use a bottom-up methodology based on the relationship:

$$E(i) = \sum_{j,k} C(j,k).EF(i,j,k).CE(j,k) \tag{1}$$

where i, j, k represents the pollutant, fuel and sector, respectively. E represents the emission of pollutant (i), EF is the emission
factor (g of pollutant per kg of burned fuel), CE is the efficiency of combustion and C is the annual fuel consumption in kilotons (kt). Note that this methodology follows the work of Junker and Liousse (2008) and Liousse et al. (2014). Details regarding improvements in the representation of sectors, emission factors, etc. are given in the following sections. In addition, two new main emissions sources have been addressed, namely flaring and open solid waste burning as detailed below.

### 2.1 Method for biofuel (BF) and fossil fuel (FF) emissions

### 2.1.1 BF and FF consumption database for 1990-2015

Fuel consumption (FC) databases used for this regional inventory are obtained from three sources: (a) United Nation Energy Statistics (UNSTAT) database, (b) International Energy Agency (IEA) database, (c) local authorities in African countries. The UNSTAT fuel consumption database for African countries provides details for 54 countries for the years 1990 to 2015. This FC database contains 22 different fuels and is available by country, fuel and sector. The IEA fuel consumption database
provides statistics for 28 African countries (all other countries (26) are gathered together) by fuel type and sector as in the UNSTAT database. The data originate from similar sources such as country reports, even though consumption totals are often different (cf supplementary documents S1). The National FC database is issued by SIE (Système d'Informations Energetiques) which is a regional organization based on UEMOA (Union Economique et Monétaire Ouest Africain) countries that collects energy data from national organizations of these countries. Each year, SIE provides an annual energy statistics report that
provides a comprehensive overview of the current energy situation in each country, as well as its evolution during the past years.

The FC dataset was first analyzed in detail country by country from 1990 to 2015 using the different data sources above mentioned. We found that the SIE values are on the same order of magnitude as those of IEA and UNSTAT databases, but incomplete. Consequently, the present work inventories are based on UNSTAT database which is the most complete. For cases
in which there is a discontinuity in the time series for a particular country (e.g. unexplained jumps in the FC trend, missing years, etc.), the data are complemented by the IEA or SIE database when available.

FC data are then grouped into 5 sectors: residential combustion sources (wood, charcoal, charcoal making, etc.), industrial, power plant, traffic and other sectors including commercial, agricultural and forestry machinery, etc. Note that details are retained for the four sub-sectors within the traffic sector (road, rail, domestic navigation and aviation).



The UNSTAT fossil fuel (FF) consumption database does not mention two-wheel (TW) vehicle consumption specifically, however this is a common and highly polluting source typical in Africa. Indeed, previous work have shown that TW vehicles use mainly smuggled fuel, particularly in the neighboring countries of Nigeria's first African crude oil exporter (Assamoi and Liousse, 2010). Our estimation of the number of TW vehicles and its fuel consumption is based on the work of Assamoi and Liousse (2010). Assamoi and Liousse (2010) estimated the number of TWs for 16 countries in West and Central Africa for the

year 2002 and 2005. This database was completed using the Demographic Health Surveys reports (DHS) (Corsi et al., 2012) statistical data for 9 countries with non-negligible TW numbers (i.e. when the consumption of the TW fleet is more than one-tenth of the country's gasoline consumption) and for which Assamoi and Liousse (2010) do not provide data. Finally, we estimate TW number per year for the entire study period (1990-2015), based on linear extrapolation techniques, using Assamoi and Liousse (2010) and DHS values. Fuel consumption for TW is also calculated based on Assamoi and Liousse (2010). For

the whole period 1990-2015, we use the mean values obtained from minimum and maximum assumptions given by this paper for TW characteristics (number of traffic days, daily consumption and fuel density) for taxis and private use only (table 1). Indeed, it has been seen that TW are used for public transportation in addition to private use in 6 west and central African countries (Benin, Cameroon, Chad, Niger, Nigeria and Togo).

### 2.1.2 Emission factors (EF)

Emission factors are dependent on fuel-type, activity, technology and norms. However, in Africa, information on technology and norms is not available for each fuel/activity and country. To deal with this limitation, our methodology is based on a "lumping" procedure designed to manage available experimental data and account for the main factors of variability. Technologies and norms are assumed to be country dependent and all 54 African countries are classified into two groups, developing (1) and semi-developed countries (2), based on their gross national product per capita (GDP) (World Bank, 2005),

for which different EFs are assigned. Twelve African countries are considered to be semi-developed countries including South Africa, Swaziland, Morocco, Algeria, etc. and the other 42 countries are classified as developing countries. Table 2 presents EF values for black carbon (BC), primary organic carbon (OC) and gaseous compounds, namely carbon monoxide (CO), non-methane volatile organic compounds (NMVOC), nitrogen oxides (NOx) and sulfur dioxide ($SO_2$) for each country type and the main anthropogenic sources.

BC and OC EFs for motor gasoline, diesel oil, two-wheel vehicles, wood burning, charcoal burning and making are taken from Keita et al. (2018) for developing countries. For semi-developed countries, the ratios of EFs between semi-developed and developing countries in Africa for specific sources given by Liousse et al. (2014) are used to estimate the EFs for this inventory. BC and OC EFs for the other sources (e.g. industries, energy, etc), as well as EFs for the other species (CO, NOx, NMVOC, $SO_2$) are provided by Liousse et al. (2014), with the exception of the non-road traffic sub-sectors. For rail and domestic

navigation using aviation gasoline (AV) and jet fuel (JF), EFs for developing countries are taken to be the highest value found in the literature as presented in Table 2. For these sub-sectors, the same EF is used for semi-developed and developing countries.





We note that, as reported in Keita et al. (2018), new EF values for BC and OC for motor gasoline, diesel oil, two-wheel vehicles, wood burning and charcoal making are slightly higher than those reported by Liousse et al. (2014), except in the case

of charcoal burning.

## 2.2 Method for open waste burning (WB) emissions

An open waste burning emission inventory is built following the IPCC guidelines (Chapters 2 and 5) for the estimation of Greenhouse Gases (GHG) that includes open residential and dump waste burning. This inventory does not include the waste burning practices in incinerators or modern combustion systems which are already accounted for in the industrial sectors.

Open waste burning is estimated using the following expression:

$$E_i = WB * EF_i ,  \tag{2}$$

where WB is the amount of solid waste that is burned residentially and in uncontrolled dump and $EF_i$ is the emission factor of pollutant i. EF values for BC and OC are provided by Keita et al. (2018) and from Akagi et al. (2011) and Wiedinmyer et al. (2014) for the other species ($NO_x$, NMVOC, CO and $SO_2$).

For each African country, the WB amount is estimated following Section 5.3.2 of the IPCC Guidelines for National GHG Inventories, 2006 which states,

$$WB = P \times MSWp \times Pfrac \ x \ Bfrac  \tag{3}$$

where P is the national population given by the World Bank database (http://data.worldbank.org/indicator, accessed 02 November 2016) and MSWp is the mass of annual per capita waste production taken from Wiedinmyer et al., (2014). The

default value recommended by IPCC of 0.6 is used for Bfrac which is the fraction of waste available to be burned that is actually burned. Pfrac is the fraction of the population assumed to burn some of their waste either near their residence or in uncontrolled dumps. Values for Pfrac used in our calculations are taken from Wiedinmyer et al., (2014). Pfrac is assumed to be based on national income status, urban versus rural population, and waste collection practices. In Africa, Pfrac may be considered to be 100% following Wiedinmyer et al. (2014). This value is the value obtained for countries with low income,

low middle income, and upper middle income, following the classification of World Bank (http://data.worldbank.org/country). In this context, semi-developed countries are not distinguished. Therefore, an overestimation of waste burning practices may be expected in some countries (e.g. South Africa, Morocco, Egypt). However, currently no data exists to take this problem in consideration.

Note that it is possible to distinguish waste burning emissions which occur near the residence to those in uncontrolled dumps.

Such practices are highly different in rural and in urban areas. As such, WB may be calculated as follows,

$$WB = WBR + WBU,  \tag{4}$$

where WBR and WBU represent waste burning in rural and urban areas, respectively, and are defined as follows,

$$WBR = MSWp \times Bfrac \ x \ Prural \times (PfracRes)R + MSWp \times Bfrac \ x \ Prural \times (PfracDump)R  \tag{5}$$

$$WBU = MSWp \times Bfrac \ x \ Purban \times (PfracRes)U + MSWp \times Bfrac \ x \ Prural \times (PfracDump)U  \tag{6}$$



In Africa, the rural population is assumed to have no organized waste collection, therefore in rural areas, (PfracRes)R linked to residential burning is assumed to be equal to 100% whereas (PfracDump)R linked to uncontrolled dumps is 0%. In urban areas, (PfracDump)U is country-dependent: for example, the fraction of uncollected waste is 0.77, 0.30, 0.40, and 0.76 for Benin, Ivory Coast, Ghana and Nigeria, respectively (Wiedinmyer et al., 2014). Therefore, in Benin for example (PfracRes)U is assumed to be equal to 77% and (PfracDump)U is 23%. Rural (Prural) and urban (Purban) populations are provided by the

World Bank database (http://data.worldbank.org/indicator, accessed 02 November 2016). Rural/urban distinction for estimating WB emissions are an important improvement to our inventory, providing details at the local and regional level which are missing in global inventories.

### 2.3 Method for emission spatial distribution for fossil fuel, biofuel and waste burning sources

The final step in the inventory construction is the disaggregation of the country-level emission totals to the African gridded

domain at 0.1 ° × 0.1 ° latitude-longitude resolution using appropriate spatial proxies for each sector. Three types of geographical information system (GIS) datasets at a resolution of 0.1° x 0.1° are used here for disaggregation: (1) 2010 population density given by CIESIN (Gridded Population of the World Future Estimate: GPWFE); (2) African country road networks based on gridded emission files from EDGAR traffic inventory (Janssens-Maenhout et al., 2011); (3) African power plant networks given by Africa infrastructure, (2009). These spatial allocation keys are used as followed: (1) road networks

for road traffic emissions, (2) geographical coordinates of power plants for energy production emissions, (3) population density grid for residential combustion sources, industries and waste burning. In the future, we plan to use gridded rural and urban population densities to better disaggregate waste burning and residential emissions over Africa.

### 2.4 Method for flaring emissions

Flaring emissions are taken from the inventory developed by Doumbia et al. (2019) for the years 1994 to 2015 using the

following equation:

$$EXflaring = GFvolume * XEF * df \tag{7}$$

where EXflaring is the emission rate of a pollutant X (kiloton) and GFvolume is the gas flare volumes in billions of cubic meters (bcm). Yearly GFvolume by country for the period 1994-2010 is taken from the National Oceanic and Atmospheric Administration (NOAA) (http://ngdc.noaa.gov/eog/dmsp/) DMSP (Defense Meteorological Satellite Program) dataset. Note

that no data exist for the years 1990-1993. For the period 2012-2015, estimations of GFvolume are based on VIRRS satellite data (Elvidge et al., 2015) and are spatially distributed based on the DMSP 2011 product. GFvolume for 2011 is estimated from trends in (1994-2010) and (2012-2015) time series.

XEF is the emission factor (EF) for species X in g/kg of fuel burned and df is the density of the fuel gas. Typically, the density of the fuel (natural gas) varies between 0.75 and 1.2 kg/m$^3$ depending on the fraction of heavy hydrocarbons present in the fuel

(US Standard Atmosphere, 1976). In this inventory, we assume a gas density of 1.0 kg/m$^3$ for converting volume of associated



gas to mass (E&P Forum, 1994). EFs for various species (CO, NOx, NMVOC, SO₂, OC and BC) are detailed by Doumbia et al. (2019) which show a large range of uncertainties. We use here the mean EF value given in this paper.

Spatial disaggregation involves overlapping layers of maps including DMSP night-time lights images, Gas flare areas, World maritime boundaries and Total gas flares volume per country based on Geographic Information System (ArcGIS software).

The final layer is gridded at 0.1° x 0.1° horizontal resolution.

## 3 Results

### 3.1 Temporal trends of African emissions

Figure 1 shows the trend of BC emissions for fossil fuel (FF), biofuel (BF), open waste burning (WB) and flaring sources in Africa from 1990 to 2015. There is an increase of 46 %, 67 % and 43 % of BC emissions for fossil fuel (FF), biofuel (BF) and

open waste burning (WB), respectively, over the time period. This is mainly due to anthropogenic activity increases linked to population growth. Indeed, Africa's population has grown by 2.5 % per year over the past 20 years, corresponding to a roughly 64 % increase over the period 1990-2015, which is of the same order of magnitude as the increase in biofuel emissions. Biofuel emissions have increased at a higher rate due to an increase in low-income population in sub-Saharan Africa where biomass constitutes about 80% of their total energy consumption (Ozturk and Bilgili, 2015).

Unlike FF, BF and WB, BC emissions from flaring are decreasing (55% from 1994 to 2015) due to the actions of the Global Gas Flaring Reduction Initiative (GGFR). In 1994, BF, FF, WB and flaring contributions to total anthropogenic BC emissions were roughly of the order of 47%, 11%, 36% and 6% whereas in 2015, such values are 45%, 17%, 35% and 2%, respectively. Increases in the contribution from FF can be explained by increases in the traffic fleet and industrialization.

Figure 2 shows the trends of pollutant emissions in five subregions of Africa (North, East, West, Middle and Southern Africa)

for the main sources (fossils fuels, biofuels and waste burning) during the period 1990-2015. The list of countries included in these five regions is given in supplementary documents table S1. All pollutant emissions are increasing globally in Africa over the period 1990-2015 at a growth rate of 95%, 86%, 113%, 112%, 97%, and 130% for BC, OC, NOx, CO, SO₂ and NMVOC compared to 1990, respectively. Except for OC, emissions of all pollutants have nearly doubled from 1990 to 2015.

BC, OC, CO and NMVOC show similar patterns of regional contributions with the highest values for West Africa, East Africa

and North Africa (except for OC, for which Middle Africa emissions are higher than North Africa emissions). Southern Africa emits more SO₂ and NOx than all the other African regions, followed by North Africa. This is due to their large amount of industrial activities compare to the other regions of Africa.

Analysis of BC emissions by region indicate that the highest emitting region is West Africa with 0.18 Tg (26%) in 1990 and 0.39 Tg (29%) in 2015 followed by East Africa with 0.17 Tg (26%) in 1990 and 0.33 Tg (25%) in 2015, North Africa with

0.15 Tg (22%) in 1990 and 0.28 Tg (21%) in 2015, Southern Africa with 0.10 Tg (14%) in 1990 and 0.17 Tg (13%) in 2015 and Middle Africa with 0.08 Tg (12%) in 1990 and 0.16 Tg (12%) in 2015. West Africa's contribution to BC emissions shows the fastest growth (26% to 29%) compared to the other regions. The highest rate of increase in BC emissions is also observed



in West Africa (117%) and the lowest in Southern Africa (70%) over the period 1990-2015. This could be in part explained by the fact that the population growth rate in Africa is higher in West Africa 2.66% and lower in Southern Africa 1.64%, as

shown by the United Nations, 2015. Furthermore, differences in BC emissions between Southern and Northern Africa regions with similar fuel consumption (for fossil fuels, biofuels and solid wastes) are partly explained by the emission factors which are higher in Northern Africa where there are more developing countries (4 over 7) than in Southern Africa (1 over 5).

In terms of $SO_2$ emissions, the highest emitting region is Southern Africa, the most industrialized region in Africa, with emissions of 1.19 Tg (73%) in 1990 and 2.24 Tg (63%) in 2015. Southern Africa is followed by North Africa, East Africa,

West Africa and Middle Africa, respectively. Over the 1990-2015 period, the highest rate of increase in $SO_2$ emissions occurred in West Africa (224%) followed by Middle Africa (71%), East Africa (56%), Southern Africa (47%) and North Africa (33%), respectively. As for BC, high rates of increase in $SO_2$ emissions during this period are observed in regions where the population growth rate is the highest, i.e. West Africa (2.66%), Middle Africa (3.10%) and East Africa (2.71%) and lower rates are found where the population growth rate is lower i.e. Southern Africa (1.64%) and North Africa (1.87%).

**3.2 Focus on the year 2015**

Figure 3 shows the spatial distribution of BC and NOx emissions in 2015 for the fossil fuel, biofuel, waste burning and flaring sources. As given in Table 3 which shows the contribution of each of these sources to the total emissions for BC, OC, NOx, $SO_2$, CO and NMVOC in 2015, total BC and NOx emissions are 1.35 Tg C and 7.90 Tg NO2, respectively. Emission densities are generally the highest along the Gulf of Guinea, and in the East and South of Africa.

Table 3 also underlines that for the whole of Africa, residential combustion is the major source of carbonaceous particles for BC (40%) and OC (77%). It is also the main source of CO and NMVOC with a contribution of 72% and 53%, respectively. Open waste burning is the second most important source of BC, OC and NMVOC representing 35%, 15% and 22% of the total, respectively. The energy sector is the major source of $SO_2$ and NOx emissions which constitute 54% and 29% of the total, respectively. For NOx emissions, the energy sector is followed by the traffic sector (26%) whereas industry is the second

largest contributor to $SO_2$ emissions (32%).

The regional contribution from each sector is presented in Fig.4. The residential sector contributes to more than 50% of BC emissions in East and West Africa, and just under 50% in Middle Africa (Figure 4a) for the year 2015. In Southern and Northern African it contributes to less than 25% and 10% of the total BC emissions, respectively. In these two regions, waste burning is the largest source of BC emissions with significant contributions from the industry and traffic sources compared to

East and West Africa. Waste burning is the second largest source of BC emissions in East and West Africa. For NOx, the traffic sector is the largest contributor in West and North Africa, making up 30% and 41% of total emissions, respectively (Figure 4b). In East Africa, the residential sector (32%) is the largest contributor of NOx emissions, followed by the traffic sector (23%). In Southern Africa, the two largest contributors to NOx emissions are the energy (52%) and industry (27%) sectors, respectively.



Figure 5 shows the sectoral contributions for specific countries in Southern West Africa (SWA) and for South Africa. As previously mentioned, for countries in SWA emissions are dominated by the residential sector, followed by open solid waste burning and the other sources, whose relative importance differs depending on the country. Note that in Nigeria, industry and flaring BC emissions are much more important compared to other countries in SWA. Also, it is interesting to show that traffic is the largest contributor to BC emissions in Benin, which can be explained by the predominance of two-wheel vehicles which

are more polluting than four-wheels vehicles. For South Africa, waste burning is the predominant sector contributing 42% to the total, (18% residentially and 24% at dump), followed by the residential (24%), industry (24%), energy (13%), traffic (9%) and flaring sectors. In Côte d'Ivoire for example, the residential sector is the most important (58%), followed by waste burning (26% with 16% residentially and 10% at dump), traffic (9%), other sector (5%), industry (1%) and energy (0.1%). We highlight that in South Africa BC emissions from the waste burning sector are higher at the dump than at residences compared to Côte

d'Ivoire where there is less organized waste collection. For $SO_2$ emissions in Africa, South Africa is the highest emitting country contributing with roughly 62% of total $SO_2$ emissions.

Finally, it is interesting to assess the relative importance of the main fuels in the emissions by sector. For BC emissions, emissions related to diesel are 7 times and 8 times higher than gasoline emissions in Côte d'Ivoire and South Africa, respectively. Similarly, wood emits about 3 times more than charcoal burning and making in Cote d'Ivoire. $SO_2$ emissions

originate mainly from the use of coal which constitutes 95% of the emissions in South Africa (51% from power plants, 31% from industry, 4% from the residential sector and 3% from other sectors). This demonstrates that actions to substitute diesel by gasoline and charcoal by wood in Côte d'Ivoire would be beneficial to reduce BC emissions. In addition, replacing coal with other fuels such as natural gas would also help reduce BC and $SO_2$ emissions in South Africa.

**3.3 Comparison with previous emission inventories**

We first compare emissions from this inventory to the African emission inventory developed by Liousse et al. (2014) which is only available for the year 2005. Table 4 summarizes emissions of pollutants (BC, OC, NOx, $SO_2$, CO and NMVOC) for the year 2005 for both inventories for the same sources. Fossil fuel and biofuel emissions from this inventory are slightly lower than those given by Liousse et al., (2014) for BC (0.64 instead of 0.69 Tg), $SO_2$ (2.53 instead of 4.02 Tg), NOx (5.08 instead of 5.80 Tg) and NMVOC (8.33 instead of 8.60 Tg), and slightly higher for OC (4.96 instead of 3.95 Tg) and CO (64.43 instead

of 58.6 Tg). These differences may be explained by the updated fuel consumption data and the new emissions factors which are taken from direct measurements from the sources in Africa. Note that this new inventory also includes two major sources not considered in the Liousse et al. (2014) inventory, namely open solid waste burning and flaring sources. As can be seen in the Table 4, including these sources increases emissions by 36%, 33%, 16% and 24% for BC, OC, CO, and NMVOC, respectively, as compared to the Liousse et al. (2014) values.

We also compare our inventory for Africa to emissions from the following global inventories : ECLIPSE V5a (Klimont et al., 2017), EDGARv4.3 (Janssens-Maenhout et al., 2011) and CEDS (Hoesly et al., 2017). These global inventories are also





developed using a bottom-up methodology, where emissions are calculated as the product of IEA activity data and emission factors for combustion sources taking into account details in technology and emission controls.

Figure 6 shows BC and NOx emission trends from our inventory and the three global inventories mentioned above. Emission values are indicated by the single lines in Figure 6 except for ECLIPSEV5a where values are indicated by dots every five years. In terms of magnitude, there are large differences among the inventories for BC, while trends are similar. BC emissions from CEDS and ECLIPSEV5a are both slightly higher than those calculated in this work (on average 12%) over the period 1990-2014. Contrarily, BC emissions from EDGARv4.3 are much lower, with emissions from the DACCIWA inventory up to 37% higher on average over the period 1990-2010. NOx emissions from the DACCIWA inventory are of the same order of magnitude as CEDS over the period 1990-2009 (difference is of the order of 3%). This difference becomes larger after 2010 where DACCIWA's NOx emissions are about 15% higher than CEDS. As for BC, DACCIWA's NOx emissions are higher on average by 18% compared to EDGARv4.3 over the period 1990-2010. Note that NOx emissions from ECLIPSE V5a are the lowest of all the other inventories. These differences are due to the different fuel and activity datasets used in DACCIWA (UNSTAT mainly) and the two global inventories (IEA), as well as differences in emission factor values used in the different inventories. The EFs used in the DACCIWA inventory are taken from direct measurements at the sources in Africa as described in Keita et al. (2018), whereas the EFs used in global inventories are largely based on measurements taken from other regions such as Europe and the U.S. which are often not appropriate for Africa. We have also compared OC and $SO_2$ emission trends from our inventory and the three global inventories mentioned above, as shown in Figure S2. In terms of OC emissions, the DACCIWA inventory has the highest value. This is mainly due to the value of the OC emission factor used for domestic fires (Keita et al., 2018), which is higher than that found in the literature based on measurements from other regions of the world with different wood species. In contrast to OC, $SO_2$ emissions from DACCIWA have the lowest values compared to other inventories. This could be due to differences in activity databases and also to the EF as in the case of NOx and BC emissions. A comparative analysis of sectoral BC and NOx emissions from the various emission inventories mentioned above is also performed. For this comparison, we use the year 2010 which is the most recent and common year for these inventories. Note that the Fugitive sector in CEDS and EDGAR inventories mainly consists of flaring emissions, whereas in the ECLIPSE inventory fugitive emissions are included in the Energy sector. Figure 7 shows the relative contribution from each sector to BC and NOx emissions in 2010 for the DACCIWA inventory and the three global inventories. While large disparities exist between the inventories, it is clear that the residential sector is the highest contributor to BC emissions in all of the inventories. The contribution of emissions from the waste sector varies greatly among the inventories (35%, 7% and 1% for DACCIWA, CEDS and EDGAR, respectively). Note that ECLIPSE has no waste sector. The traffic and energy sectors are the largest emitters of NOx for both the CEDS (31% and 27%, respectively) and EDGAR4.3 (34% and 32%, respectively) inventories. In the DACCIWA inventory, the contribution from the energy sector (28%) is slightly larger than from the transport sector (25%).

Finally, Liousse et al. (2014) have estimated the emissions of BC and OC in 2005 and 2030, using different scenarios. These values are shown by the dots in Figure S1 for REF and CCC* scenarios. The authors defined the REF scenario as the state of



the world from the perspective of "business and technical change as usual", driven solely by basic economics and CCC* as a scenario incorporating both the introduction of carbon penalties and African specific regulations implemented to achieve a large reduction in emissions from incomplete combustion. We have linearly extrapolated the DACCIWA emissions for these two species, as shown by the plain lines in Figure S1. Our estimates for BC and OC are higher than the best case scenario values and lower than the worse scenario values of Liousse et al. (2014). OC values are being much closer to the worse scenario and BC values closer to the best case scenario. These results demonstrate that emission mitigation measures need to be implemented urgently in Africa in order to avoid such elevated emissions in 2030.

## 3.4 Emission uncertainty analysis

### 3.4.1 Method for emission uncertainty calculation

Uncertainties in emission inventories are mainly due to the lack of information such as fuel consumption and accurate emission factors. However, the main challenges in the estimation of uncertainties in emissions are related to the uncertainties in input data and in the development of methods for quantifying systematic errors. In this study, we use a Monte Carlo statistical method to quantify uncertainties in the input data such as emission factors and fuel consumption data. Specifically, a Monte Carlo simulation is performed in order to quantify the uncertainty in emission estimates (Frey and Zheng, 2002; Frey and Li, 2003; Zhao et al., 2011). Parametric distributions and standard deviation linked to the reliability and accuracy of data introduced by fuel consumption statistics and non-national emission factors are provided in the literature (e.g. IPCC 2006; Zhao et al., 2011; Bond et al., 2004) and expert judgment. In this study, we assume that when the coefficient of variation is less than approximately 30%, the distribution is normal (IPCC, 2006). When the coefficient of variation is larger and the quantity is non-negative, an asymmetric lognormal distribution is assumed. The Monte Carlo method was then used to propagate these uncertainties to obtain the uncertainty on emissions for each fuel per sector. The Monte Carlo analysis consisted of selecting random values of activity and emission factors data from the respective distributions to obtain the corresponding emissions. This calculation was repeated 100,000 times to obtain the average value of the 100,000 emission values and their distributions. The standard deviations for these distributions were estimated with a 95% confidence interval. Uncertainty of total emission per pollutant was obtained by combining values obtained by fuel oil and by sector.

### 3.4.2 Uncertainty results

The uncertainties in the emission estimates are within [-35%; +58%], [-15%; +20%], [-17%; +21%], [-16%; +18%], [-14%; +16%] and [-17%; +23%] for BC, NOx, OC, CO, NMVOC and $SO_2$, respectively. As expected, the higher uncertainties are observed for the smallest emission values (e.g. BC). Uncertainties in biofuel activity data (20-100%) are greater than those for fossil fuels (10-20%) because of the absence of official markets for biofuel fuels and consequently lower accuracy in their consumption estimates. Biofuel used in the residential sector shows the highest uncertainties for this inventory, for example





with [-40%; +101%] and [-38%; +171%] for BC and NOx, respectively, with a 95% confidence interval. For fossil fuel, the highest uncertainties are obtained for two-wheel vehicles with [-62%; +115%] and [-74%; +173%] for BC and NOx, respectively, with 95% confidence interval. For two-wheel fuels, higher uncertainties are due to the lack of statistics on the

number of two-wheeled vehicles and its consumption. Note that some uncertainties are not taken into account. For example, EFs were considered to be constant over the studied period, while the GDP of countries used to vary as well as the composition of the TWs (2 and 4 strokes ratio). Also, waste burning emission estimates in Southern Africa assume that Southern Africa is comprised of developing countries only. Emissions are then overestimated since the parametrizations should include characteristics of both semi-developed and developing countries. This problem is not included in the uncertainty calculations.

## 4 Conclusion


Within the framework of DACCIWA, a new African emission inventory has been developed for fossil fuel, biofuel, open waste burning and gas flaring for the years 1990-2015. Emissions of BC, OC, CO, NMVOC, $SO_2$ and NOx are included for the residential, industry, energy, transportation, open waste burning, flaring and other sectors. These emissions are provided at a spatial resolution of 0.1° x 0.1°. This inventory uses new emission factor derived from direct measurements at main

emission sources in Africa, new spatial distribution proxies and the addition of new emission sources as compared to previous regional African emission inventories.

In this paper, emissions are discussed in the context of five geographical regions in Africa. Our analysis highlights differences in the characteristics of both fuel consumption and pollutant emissions in these regions. West Africa is identified as the highest emitting region of BC, OC, CO and NMVOC, followed by East Africa and/or North Africa. Southern Africa and North Africa

are the highest emitting regions of $SO_2$ and NOx. These differences are due to the relative contribution of emissions from different activity sectors. High $SO_2$ and NOx emissions in Southern Africa and North Africa are linked to their large quantities of industrial activities and thermal power plants, whereas in regions with more developing countries (44 out of 56 African countries) such as West or East Africa, higher emissions from the domestic and traffic sectors are found.

Comparisons with other inventories reveal significant differences between both regional inventories such as Liousse et al.

(2014) as well as global inventories. Differences with Liousse et al. (2014) are largely due to updated activity data and emission factors used in this inventory. The DACCIWA emissions are within the range of the three global inventories, more specifically, between EDGARD4.3 (lower bound) and CEDS and ECLIPSEV5a (upper bound), depending on the species.

In addition, an estimation of the emission uncertainties has been presented. These uncertainties are due to both the lack of reliable statistics on data from the sectors above mentioned and as well as the assumptions which have been made (e. g. constant

EFs over the study period, less well-known EFs for industry and flaring sources). Finally, this inventory highlights key pollutant emission sectors to focus on in the development of mitigation scenarios. Notably, fuel changes or implementation of less emitting technologies are suggested.



## Availability of the data


Annual gridded emissions at a spatial resolution of 0.1° x 0.1° for BC, OC, CO, NMVOC, $SO_2$ and NOx from the residential, industry, energy, traffic, open waste burning, flaring and other sectors for the years 1990 to 2015 are provided in NetCDF format (DACCIWA, https://doi.org/10.25326/56, Keita et al., 2017) and are available through the Emissions of atmospheric Compounds and Compilation of Ancillary Data (ECCAD) system with a login account (https://eccad.aeris-data.fr/). For review

purposes, ECCAD has set up an anonymous repository where subsets of the DACCIWA data can be accessed directly https://www7.obs-mip.fr/eccad/essd-surf-emis-dacciwa/.



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



**Lists of figures**



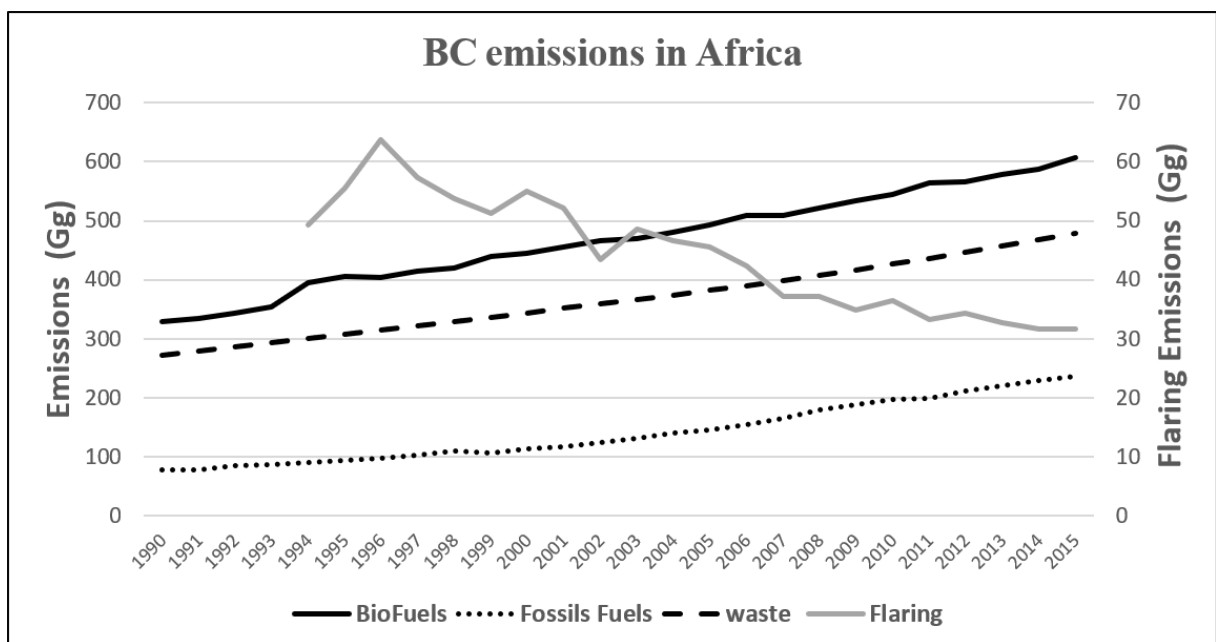

**Figure 1: Trend of BC emissions in Africa for fossils fuels, biofuels, open waste burning and flaring**







**Figure 2 : Trends in regional emission estimates of BC, OC, NOx, CO, SO₂ and NMVOC, for fossils fuels, biofuel and waste burning**
**sources.**



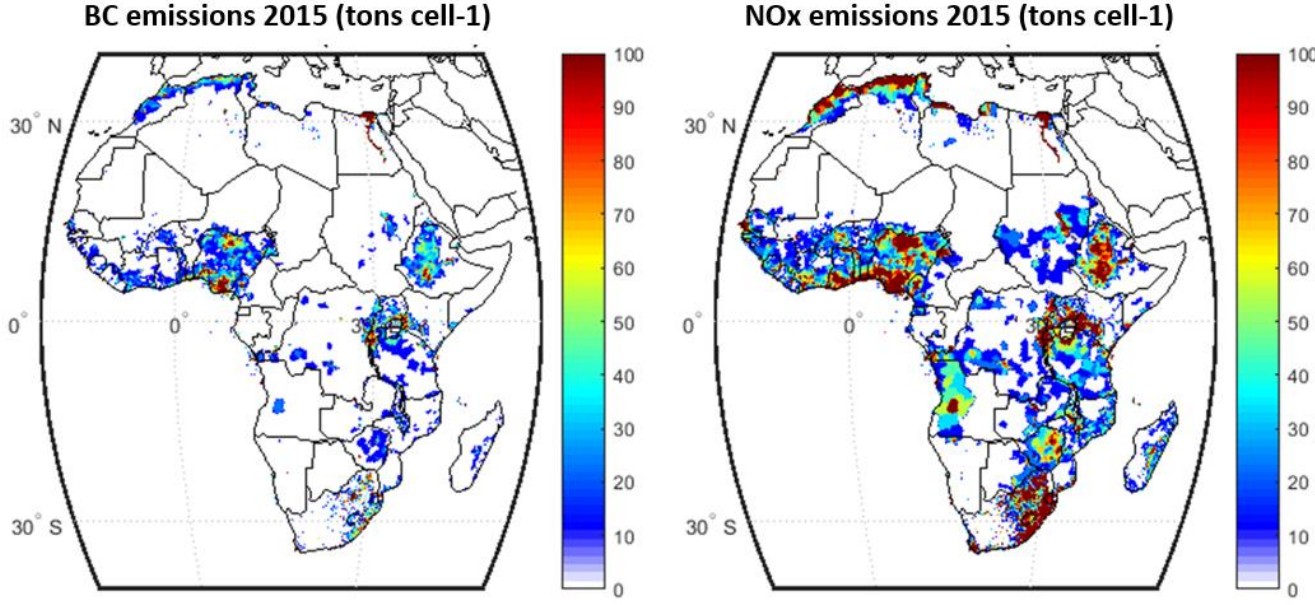

**Figure 3: Spatial distribution of total anthropogenic BC and NOx emissions in 2015**





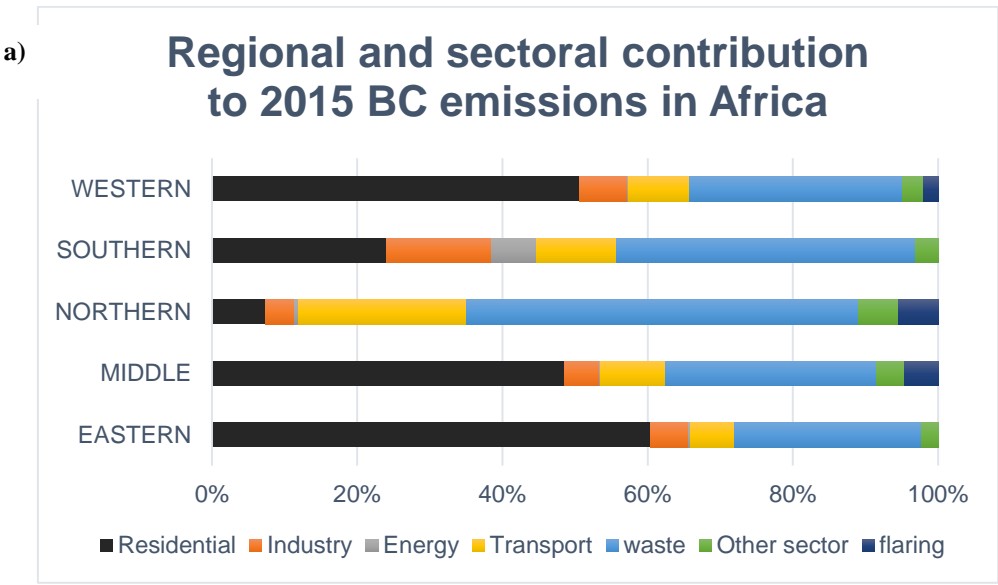

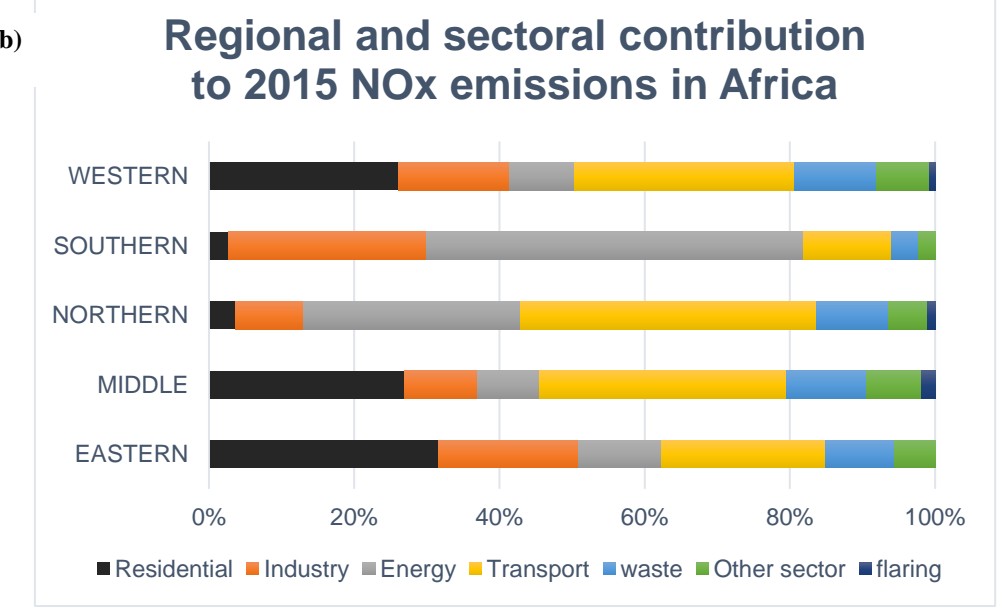

**Figure 4: Regional sectoral contribution to 2015 BC (a) and NOx (b) emissions in Africa**




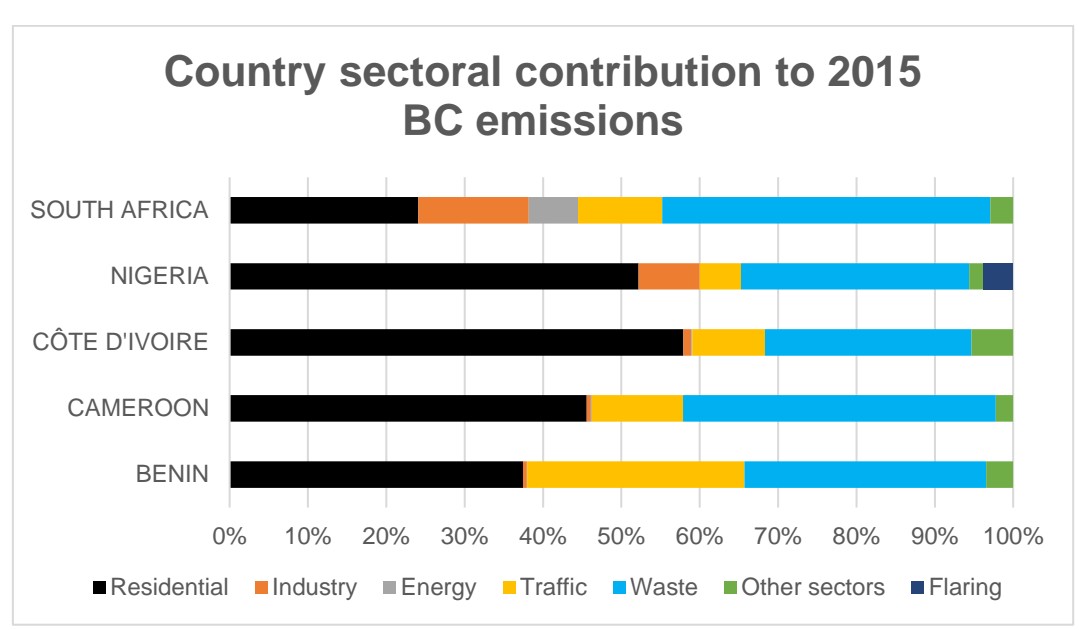

**Figure 5: Sectoral contribution to 2015 BC emissions in some African countries**





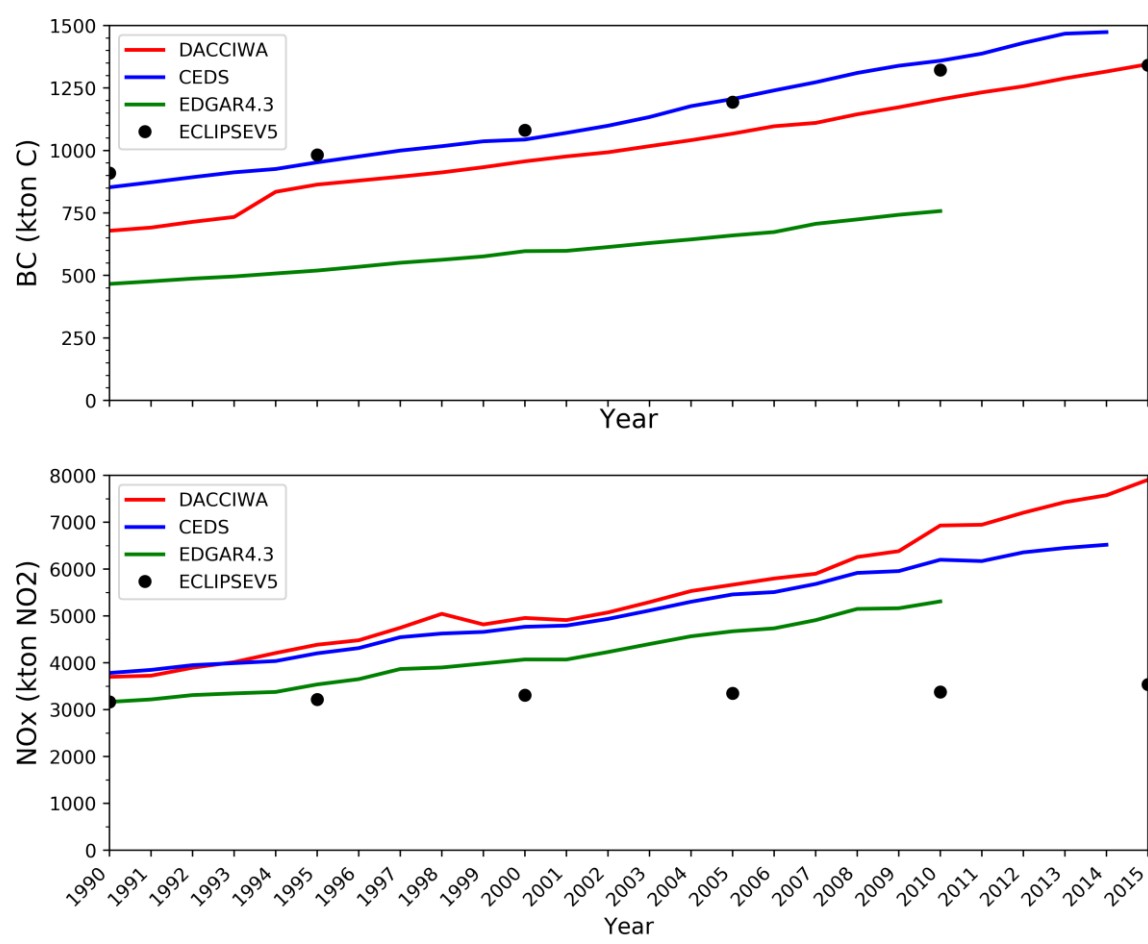

**Figure 6: Comparison of NOx (top) and BC (bottom) between DACCIWA inventory (this work) and global inventories (CEDS, EDGAR and ECLIPSE inventories).**





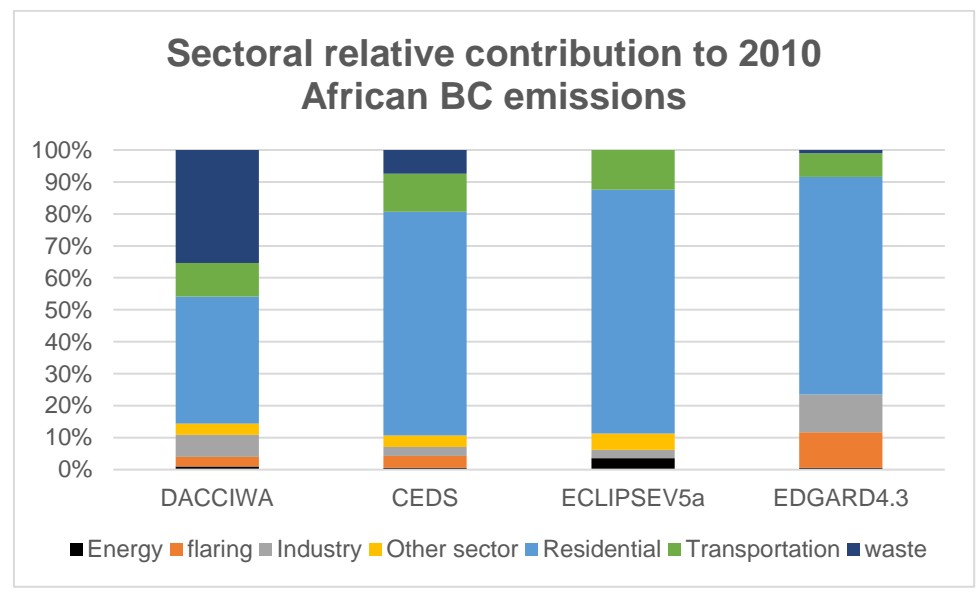

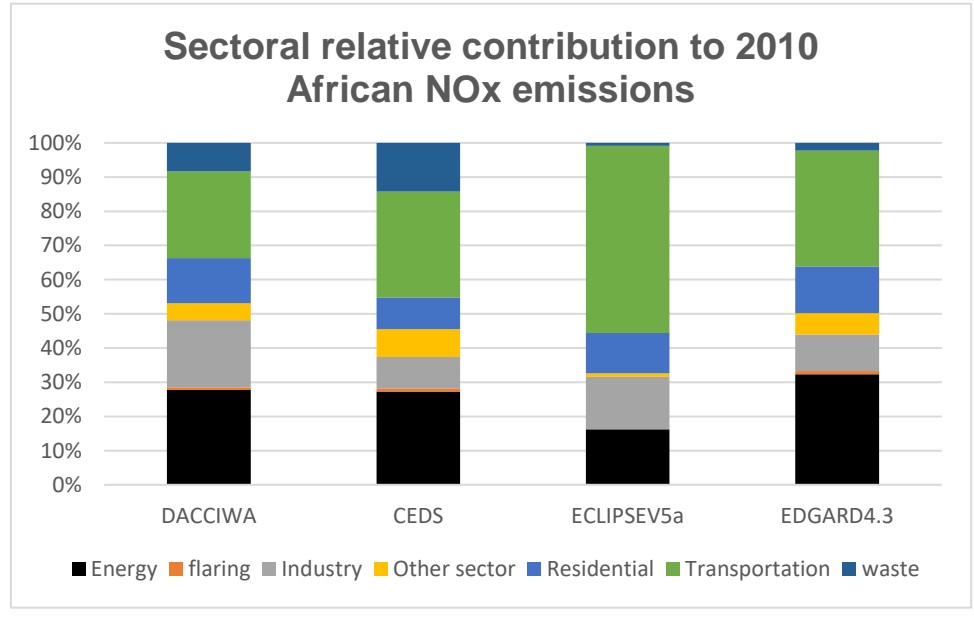


**Figure 7: Sectoral Relative contribution for BC and NOx emissions in 2010 for CEDS, ECLIPSEV5a, EDGAR4.3 and DACCIWA (this work) inventories.**





**Lists of Tables**





**Table 1: Two-wheel (TW) characteristics based on the minimum and maximum assumptions from Assamoi and Liousse (2010).**

|  | Number of traffic day(s) per week | Daily consumption (in liter) | Fuel density (in kg/m³) |
|---|---|---|---|
| TW Private use | 6 | 0.75 | $\rho = 754$ |
| TW-taxis | 6 | 1.875 | $\rho = 754$ |









**Table 2: BC, OC, CO, NOx, SO₂ and NMVOC EFs for the main anthropogenic sources, for different fuels (AV: aviation gasoline, JF: jet fuel, DL: diesel, MO: motor gasoline, RF: residual fuel oil, FW: wood, CH: charcoal and CHM: charcoal making), type of country (1: semi-developed, 2: developing) and activities sectors (DAV: domestic aviation, DNAV: domestic navigation, RAIL: rail traffic, ROAD: road traffic and D: residential combustion).**

| Fuel/country | Sector | BC | OC | CO | NOx | SO₂ | NMVOC |
|---|---|---|---|---|---|---|---|
| | | gC/kg (dm) | gC/kg (dm) | gCO/kg (dm) | gNO₂/kg (dm) | gSO₂/kg (dm) | gNMVOC/kg (dm) |
| AV/1/2 | DAV | 0.1[a] | 0.025[a] | 8.265[f] | 11.5[f] | 0.97[f] | 1.88[f] |
| JF/1/2 | DAV | 0.1[a] | 0.025[a] | 8.15[b] | 10.18[b] | 0.98[b] | 0.353[b] |
| DL/1 | DNAV | 1.318[h] | 0.926[h] | 7.4[g] | 78.5[g] | 0.04[g] | 2.8[g]/3[a] |
| DL/1/2 | RAIL | 1[e]/1.34[h] | 0.72[e]/0.75[h] | 10.8[e] | 48.3[e]/52.4[d] | 0.02[e] | 4[a]/4.65[d] |
| DL/1/2 | ROAD | 4.47[g]/2.0[g] | 3.53[g]/1.0[g] | 37[c]/14.8[c] | 34.4[c]/13.76[c] | 0.72[c]/0.29[c] | 3.04[c]/3.04[c] |
| MO/1/2 | ROAD | 0.52[g]/0.15[g] | 0.906[g] | 300[c]/300[c] | 19.5[c]/19.5[c] | 2.36[c]/2.36[c] | 28.1[c]/28.1[c] |
| RF/1/2 | DNAV | 1.318[h] | 0.926[h] | 7.4[g] | 79.3[g] | 0.3[g] | 2.7[g] |
| FW/1/2 | D | 0.825[g]/0.75[g] | 9.286[g]/4.643[g] | 75.6[c]/63[c] | 1.325[c]/1.1046[c] | 0.2[c] | 8.76[c]/7.3[c] |
| CH/1/2 | D | 0.65[g] | 1.78[g] | 200[c] | 5.967[c] | 0.4[c] | 4.87[c] |
| CHM/1/2 | D | 0.15[g] | 3.04[g] | 69[c] | 0.07[c] | 0.01[c] | 12[c] |

[a] Wei et al., (2008)

[b] Kurniawan and Khardi, (2011)

[c] Liousse et al., 2014

[d] eea/1.A.3.C Railways/tier 1

[e] TRANSFORM, deliverable D1.2.5, type Report on railway emission factor

[f] IPCC, Reference Manuel (Revised, 2007)

[g] Keita et al., 2018

[h] IIASA, GAIN EF

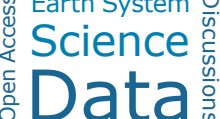

**Table 3: Sectoral emissions of carbonaceous particles and combustion gases in 2015, in Africa Gg. year-1**

| 2015 (Gg) | BC | OC | NOx | SO₂ | CO | NMVOC |
|---|---|---|---|---|---|---|
| Residential | 539.9 | 5706.7 | 1043.3 | 225.4 | 68056.3 | 8393.8 |
| Industry | 88.2 | 213.5 | 1402.2 | 1135.2 | 1296.3 | 197.1 |
| Energy | 13.4 | 14.7 | 2292.4 | 1906.1 | 403.2 | 23.2 |
| Traffic | 155.1 | 313.0 | 2088.6 | 118.4 | 15783.6 | 3356.8 |
| Waste | 478.3 | 1109.0 | 644.1 | 86.1 | 6544.0 | 3520.0 |
| Other sectors | 46.8 | 67.6 | 390.4 | 82.2 | 2608.0 | 241.8 |
| Flaring | 31.7 | 3.7 | 44.8 | 1.7 | 196.9 | 134.1 |
| Total anthropogenic | **1353.4** | **7428.2** | **7905.8** | **3555.1** | **94888.3** | **15866.8** |








**Table 4: Comparison of fossil fuel and biofuel emissions between this work and Liousse et al. (2014) inventory for the year 2005.**

| | | \multicolumn{6}{c}{Emissions for the year 2005 (Tg)} | | | | | |
| Sources | Inventory | BC | OC | CO | NMVOC | NOx | SO$_2$ |
|---|---|---|---|---|---|---|---|
| Fossils Fuels + BioFuels | This work | 0.64 | 4.97 | 64.43 | 8.33 | 5.08 | 2.53 |
| | Liousse et al., 2014 | 0.69 | 3.96 | 58.61 | 8.57 | 5.81 | 4.02 |
| Fossils Fuels + BioFuels + Waste burning + Flaring | This work | 1.07 | 5.92 | 69.95 | 11.34 | 5.66 | 2.60 |




