# Peer review of "African Anthropogenic Emissions Inventory for gases and particles from 1990 to 2015"

_Earth System Science Data, 2020_

## Referee Comment (RC1) · Anonymous Referee #1 · 8 Mar 2021

**Review of "African Anthropogenic Emissions Inventory for gases and particles from 1990 to 2015" by S. Keita et al. submitted to ESSD in 2020**

**General Description:**
The authors develop and describe a long-term anthropogenic emission inventory for Africa spanning 1990-2015. They also compare it to other regional inventories as well global inventories commonly used in models and estimate errors in the inventory using Monte Carlo simulations. The data are also available for public use in an accessible data repository. This is an important development for a continent that is experiencing rapid growth and severe degradation in air quality and that is often poorly represented in global inventories. Prior to publication in ESSD, the authors need to make their description of the methods clearer and also compare their emissions estimates to published regional inventories developed by other research groups. There are also many typos, grammatical errors, unfamiliar acronyms, and unclear statements in the manuscript. These are detailed below.

**General Comments:**
The authors only compare to regional inventories that developed by the same group. What about others developed for the continent and target African countries? These include, but are likely not limited to, Marais and Wiedinmyer (2016) for multiple inefficient and diffuse combustion sources across Africa, Bockarie et al. (2020) for charcoal production and use in Africa, Pretorius et al. (2015) for power stations in South Africa.

At the end of reading the paper, it isn't apparent whether this is a better inventory (more representative of conditions in Africa) than those the authors compare to.

There are many typos and formatting issues throughout the manuscript. These include, but may not be limited to:
- Unnecessary commas on line 33 "2050,." and in many in-text citations in the format "Author et al.**,** (YYYY)"
- Define acronyms on first use. For example, POLCA (line 35) and DACCIWA (line 38). Not all readers will know what these are.
- VIIRS is misspelled (line 185)
- Mathematical symbols used in the equations are inconsistent. The equations use ".", "*", and "×" for multiplication. Use "×" throughout for accuracy and consistency.
- Throughout, "x" in $NO_x$ should be subscripted. This is also the case for "2" in $NO_2$ and "2.5" in $PM_{2.5}$.
- Throughout, change NMVOC to plural, NMVOCs, as presumably there is more than one represented in the inventory.

Often "growth rate" is used, but then a percentage increase value spanning 1990-2015 is given. If it's a growth rate, it would be in units of "% per year or decade". In many instances in the manuscript (e.g., lines 212 and 230) it would be more appropriate to refer to these as "growth" or "increase".

Are Eqs (4)-(6) necessary? It seems that the information in the equations could be more clearly expressed in words rather than using obscure and unfamiliar acronyms that make it hard to follow along.

Many unfamiliar acronyms are used. In many instances these could just be written in full (e.g., TW for two-wheel). This will also make it easier for the reader to follow along.

Figures in the supplementary include acronyms with no explanatory text (proj, ccc, ref). Define these in the caption so that the reader doesn't need to search for these in the manuscript.

**Specific Comments:**

Abstract, line 24: give the finding from the comparison.

Line 51: Name some of these important missing sources.

Line 54: Change "the entire of Africa" to "Africa" or "the whole continent".

Line 57: The longest period compared to what? There are global inventories that span longer time periods (McDuffie et al., 2020).

Line 68 (Equation (1)): What values do you use for the combustion efficiency (CE)?

Lines 76-78: Provide DOIs or URLs for the UNSTAT and IEA data sources.

lines 80: What does "gathered together" mean? Does this mean that the data for these are reported by the IEA as a single summed value? How much do these 26 countries contribute to the total?

Line 82: What does this point to in the supplementary document? There doesn't appear to be any consumption data in the supplementary material.

Line 94 and Table 1: What is "domestic navigation"?

Lines 95-108: Either change the acronym to two-wheel vehicles (TWV) or include "vehicles" after TW where relevant, otherwise sentences are incomplete. For example, "Fuel

consumption for TW is also calculated ..." reads as "Fuel consumption for two-wheel is also calculated ...". There are other entries in this paragraph that are also incomplete.

Line 97: What does "Nigeria's first African crude oil exporter" refer to?

Lines 110-113: What does "norms" mean? Consider rewording so that it is clear what this refers to.

Line 116: Swaziland is now eSwatini.

Lines 125-126: Would the highest value always be appropriate? The highest value for $NO_x$ assumes the highest combustion efficiency, whereas the highest values for OC, NMVOCs and CO assume the lowest combustion efficiency.

Line 129: What is the "slightly higher" value? Quote the number.

Line 152: Why these countries? Because of their development status?

Line 166: Important improvement relative to what?

Lines 171-172: Why use only 2010 population distribution? Won't this misrepresent population distribution, in particular in urban areas where urbanization rates are high?

Line 173: What data does EDGAR use to map traffic in Africa?

Line 174: The "Africa infrastructure (2009)" reference doesn't appear to be in the reference list.

Lines 176-177: How about also considering improved mapping of charcoal production in rural areas, as in Bockarie et al. (2020)?

Line 186: Briefly tell the reader why gas flare volume values are missing for 2011.

Line 188: "df" already has a common usage (derivative). Consider using the standard notation (Greek symbol rho) for density.

Line 192: "in this paper" refers to which paper? Doumbia? If so, rather say "in Doumbia et al. (2019)".

Line 198, Figures 1 and 2: These look like a time series, not trends. There are no trend lines or values in these figures.

Line 203: Higher rate than what?

Line 211: "globally in Africa" is confusing. Which is it? Global or in Africa?

Line 227: What does "(4 over 7)" and "(1 over 5)" mean? Should "over" be "of"?

Lines 233-234: What are the numbers in brackets? Population or emission growth rates? Are the units correct? Should these be "% a$^{-1}$"?

Lines 240-241: Has the relative contribution of the different sectors changed over time?

Line 255: What's the justification for looking at these countries specifically?

Line 259: Is there a quantitative way to express the predominance of these two-wheel vehicles?

Line 265: Is it industry or coal-fired power plants in South Africa that make the largest contribution? How does your estimate compare to emissions from Pretorius et al. (2015)?

Line 275: The Liousse et al. (2015) inventory projected dramatic increases in emissions in Africa. How do the % per year increase values in that inventory compare to your inventory?

Line 278: Is $SO_2$ "slightly lower"? The difference is ~1.5 Tg. What accounts for this difference?

lines 280-281: The reasons for the difference are quite generic (could be said for any emission inventory comparison). Can you be more specific about the source and sizes of the differences in activity factors and emission factors?

Lines 285-286: Are these global emission inventories independent of each other for Africa?

Line 291: What accounts for these differences in BC and $NO_x$ (and likely other compounds too)?

Lines 299-302: Where do the other global inventories get their EFs from and by how much do these differ from the EFs used to develop your inventory?

Line 305: Higher by how much?

Line 320: Using the REF and CCC* acronyms immediately without first defining them assumes the reader knows what these are.

Section 3.4: Are subsections (3.4.1-3.4.2) is this section necessary? Each subsection is only a paragraph long.

Line 358: Would emissions always be overestimated? Is this because combustion efficiency varies with development stage? Wouldn't this lead to an overestimate for some emitted compounds and underestimate for others?

Figure 3: "-1" in "cell-1" should be superscripted.

Figure 6: Consider including a line starting from 2005 for the projected emissions from the Liousse et al. inventory.

Figure 7: What is the second "**D**" in "EDGAR**D**4.3"

Table 1: Is this table necessary, when the only difference is the daily consumption? This could be more succinctly stated in the text.

Table 2: Should "CH" be "charcoal use", rather than just "charcoal"?

Table 3: Fix "Gg. year-1".

**References:**
Bockarie et al., 2020, doi:10.1021/acs.est.0c03754.
Marais and Wiedinmyer, 2016, doi:10.1021/acs.est.6b02602.
McDuffie et al., 2020, doi:10.5194/essd-12-3413-2020.
Pretorius et al., 2015, http://www.scielo.org.za/scielo.php?script=sci_abstract&pid=S1021-447X2015000300004
Wiedinmyer et al., 2014, doi:10.1021/es502250z

---

## Referee Comment (RC2) · Anonymous Referee #2 · 12 Mar 2021

In this paper the authors provide a very valuable emission inventory for a region for which very little information is available with regard to anthropogenic emissions. This is a well-written and original paper that I must be accepted for publication in a journal such as ESSD. There are, however, a number of typographical errors throughout the paper, which must be addressed by the authors. This can be rectified through subjecting the paper for language editing by a English language editor, as well as assistance with text editing by a native English speaker with a background in atmospheric science.

---

## Author Comment (AC1) · 19 Apr 2021

**Review of "African Anthropogenic Emissions Inventory for gases and particles from 1990 to 2015" by S. Keita et al. submitted to ESSD in 2020**

By Keita et al.

**Response to Reviewer's comments**

Dear Editor,

First, we would like to thank the reviewers for their positive comments and their suggestions to improve the quality of this document. All the questions were treated and our document was fully revised taking into account all the reviewers' comments. The paper, figures and tables were modified as a result of the different suggestions and remarks. Please find attached a point-by-point response to the reviewers' questions.

We have indicated the referees' comments in black, the authors' responses in blue and the changes in the revised manuscript in red.

**Reviewer's comments**

**Referee #1**

**Anonymous Referee #1**

**General Description:**
The authors develop and describe a long-term anthropogenic emission inventory for Africa spanning 1990-2015. They also compare it to other regional inventories as well global inventories commonly used in models and estimate errors in the inventory using Monte Carlo simulations. The data are also available for public use in an accessible data repository. This is an important development for a continent that is experiencing rapid growth and severe degradation in air quality and that is often poorly represented in global inventories. Prior to publication in ESSD, the authors need to make their description of the methods clearer and also compare their emissions estimates to published regional inventories developed by other research groups. There are also many typos, grammatical errors, unfamiliar acronyms, and unclear statements in the manuscript. These are detailed below.

We thank Referee #1 for providing very useful comments and suggestions on the manuscript. Following the referee' comments, many changes were made in the manuscript, and have been taken into account following the general and specific comments.

**General Comments:**
The authors only compare to regional inventories that developed by the same group. What about others developed for the continent and target African countries? These include, but are likely not limited to, Marais and Wiedinmyer (2016) for multiple inefficient and diffuse combustion sources across Africa, Bockarie et al. (2020) for charcoal production and use in Africa, Pretorius et al. (2015) for power stations in South Africa.

As suggested by the reviewer, the comparison of our emissions with published regional and global inventories has been added to the manuscript (see section 3.3 in lines 318-390).

At the end of reading the paper, it isn't apparent whether this is a better inventory (more representative of conditions in Africa) than those the authors compare to.

We believe that this emission inventory has smaller uncertainties because it uses emission factors specific to African emissions sources

There are many typos and formatting issues throughout the manuscript. These include, but may not be limited to:
• Unnecessary commas on line 33 "2050,." and in many in-text citations in the format "Author et al.**,** (YYYY)"

Unnecessary commas on line 33 "2050,." and in many in-text citations in the format "Author et al., (YYYY)" have been deleted

• Define acronyms on first use. For example, POLCA (line 35) and DACCIWA (line 38). Not all readers will know what these are.

These acronyms have been defined when they are first used in the manuscript.

• VIIRS is misspelled (line 185)

This has been corrected

• Mathematical symbols used in the equations are inconsistent. The equations use ".", "*", and "x" for multiplication. Use "x" throughout for accuracy and consistency.

This has been corrected

• Throughout, "x" in NOx should be subscripted. This is also the case for "2" in NO2 and "2.5" in PM2.5.

This has been corrected

• Throughout, change NMVOC to plural, NMVOCs, as presumably there is more than one represented in the inventory.

This has been corrected

Often "growth rate" is used, but then a percentage increase value spanning 1990-2015 is given. If it's a growth rate, it would be in units of "% per year or decade". In many instances in the

manuscript (e.g., lines 212 and 230) it would be more appropriate to refer to these as "growth" or "increase".

This has been corrected and detailed in the responses to the specific comments.

Are Eqs (4)-(6) necessary? It seems that the information in the equations could be more clearly expressed in words rather than using obscure and unfamiliar acronyms that make it hard to follow along.

We think that Eqs (4)-(6) are necessary for a good understanding of the methodology.

Many unfamiliar acronyms are used. In many instances these could just be written in full (e.g., TW for two-wheel). This will also make it easier for the reader to follow along.

This has been taken into account in the manuscript as detailed in the answers to the specific comments.

Figures in the supplementary include acronyms with no explanatory text (proj, ccc, ref). Define these in the caption so that the reader doesn't need to search for these in the manuscript.

This has been corrected.

**The responses to the Referees comments on Specific issues are found below.**

**Specific Comments:**

Abstract, line 24: give the finding from the comparison.

This sentence was rewritten in lines 25-26 as:

"Emissions from this inventory are compared to other regional and global inventories and the emission uncertainties are quantified by a Monte Carlo simulation." The comparison between this emission inventory and others show that differences exist, ranging from 8% to 40%.

Line 51: Name some of these important missing sources.

Some of these important missing sources are cited in lines 57-59.

Details have been given on uncertainties in the text: "there is a lack of reliable statistic on national activity data and emission factor specific to the sources considered, population density is used as a default spatialization proxy for all sectors"

Line 54: Change "the entire of Africa" to "Africa" or "the whole continent".

"the entire of Africa" has been replaced by "Africa"

Line 57: The longest period compared to what? There are global inventories that span longer time periods (McDuffie et al., 2020).

This sentence has been corrected in line 66:

"this inventory covers the 1990-2015 period"

Line 68 (Equation (1)): What values do you use for the combustion efficiency (CE)?

Combustion Efficiency values used in our calculations are now detailed in lines 82-85 of the manuscript:

"Mean CE values obtained by Keita et al. (2018) and typical for a mix of smoldering and flaming combustion conditions, have been used for solid biofuels (e.g. 0.84 for fuelwood, 0.83 for charcoal use and 0.76 for charcoal making). For liquid fuels (kerosene, gasoline, diesel and liquefied petroleum gas) and natural gas, we chose CE=1."

Lines 76-78: Provide DOIs or URLs for the UNSTAT and IEA data sources.

URLs for the UNSTAT and IEA data sources have been added in lines 91-92 of the text:

UNSTAT (http://data.un.org/Explorer.aspx), IEA (https://www.iea.org/data-and-statistics/data-tables?).

lines 80: What does "gathered together" mean? Does this mean that the data for these are reported by the IEA as a single summed value? How much do these 26 countries contribute to the total?

Yes, this means that these data are reported by the IEA as a single summed value.

Line 82: What does this point to in the supplementary document? There doesn't appear to be any consumption data in the supplementary material.

We have removed the mention of the supplementary document, as S1 does not give consumption data.

Line 94 and Table 1: What is "domestic navigation"?

Domestic navigation is navigation that takes place within the boundaries of the country, it concerns fuels delivered to ships of all flags that are not engaged in international navigation.

Lines 95-108: Either change the acronym to two-wheel vehicles (TWV) or include "vehicles" after TW where relevant, otherwise sentences are incomplete. For example, "Fuel consumption for TW is also calculated ..." reads as "Fuel consumption for two-wheel is also calculated ...". There are other entries in this paragraph that are also incomplete.

We have included "vehicles" after TW where relevant in the paper

Line 97: What does "Nigeria's first African crude oil exporter" refer to?

Nigeria is Africa's largest producer and exporter of crude oil, implying a proliferation of oil smuggling to its neighbouring countries. "producer" was added in the text

Lines 110-113: What does "norms" mean? Consider rewording so that it is clear what this refers to.

Norm refers here to emissions reduction regulations, and we have changed the text. The sentence was rewording in line 128:

"Emission factors are dependent on fuel-type, activity, technology and emission reduction regulations"

Line 116: Swaziland is now eSwatini.

Swaziland was replaced by Eswatini.

Lines 125-126: Would the highest value always be appropriate? The highest value for NOx assumes the highest combustion efficiency, whereas the highest values for OC, NMVOCs and CO assume the lowest combustion efficiency.

Our study includes not only the combustion efficiency, but also the quality of fuel etc. In that case, we estimated that the choice of the highest value is more accurate for developing countries.

Line 129: What is the "slightly higher" value? Quote the number.

This sentence has been rewritten in lines 148-151:

"It should be noted that, as reported in Keita et al. (2018), new EF values for BC and OC for motor gasoline, diesel oil, and wood burning are higher than those reported by Liousse et al. (2014) (for example for motor gasoline by a factor of 1.5 and 4 respectively for OC and BC), and slightly lower in the case of charcoal burning, charcoal making and two-wheel vehicles (for example for charcoal making by a factor of 0.8 and 0.9 respectively for OC and BC)".

Line 152: Why these countries? Because of their development status?

We have changed the sentence to make it more clear in lines 175-177:

"In this context, data for semi-developed countries are not available. Therefore, an overestimation of waste burning practices may be expected in some countries (e.g. South Africa, Morocco, Egypt). However, currently no data exists to avoid this overestimation."

Line 166: Important improvement relative to what?

The improvement is relative to global emission inventory. We have modified the existing sentence in lines 193-194:
"Rural/urban distinction for estimating WB emissions are an important improvement to our inventory, providing details at the local and regional levels which are missing in global inventories."

Lines 171-172: Why use only 2010 population distribution? Won't this misrepresent population distribution, in particular in urban areas where urbanization rates are high?

When we started to develop the DACCIWA emissions, the 2015 population were "interpolated" data, and we preferred to use the 2010 real values.

Line 173: What data does EDGAR use to map traffic in Africa?

EDGAR uses a world map road traffic. We extracted a road map data from the EDGAR dataset.

Line 174: The "Africa infrastructure (2009)" reference doesn't appear to be in the reference list.

The Africa infrastructure (2009) URL has been added in the text (https://powerafrica.opendataforafrica.org/).

Lines 176-177: How about also considering improved mapping of charcoal production in rural areas, as in Bockarie et al. (2020)?

As indicated in lines 176 -177, better mapping for a few sectors will be considered in the future.

Line 186: Briefly tell the reader why gas flare volume values are missing for 2011.

This explanation has been added to the paper in lines 217-218:

"$GF_{volume}$ for 2011 is estimated from 1994-2010 trends and from the 2012-2015 time series as the 2011 DMSP data cannot be used due to an orbital degradation that led to solar contamination."

Line 188: "df" already has a common usage (derivative). Consider using the standard notation (Greek symbol rho) for density.

This has been corrected "df" has been replaced by "$\rho$"

Line 192: "in this paper" refers to which paper? Doumbia? If so, rather say "in Doumbia et al. (2019)".

"in this paper" refers to Doumbia et al. (2019). The sentence has been rewritten in lines 223-224 as:

"We use here the mean EF value given in Doumbia et al. (2019) paper"

Line 198, Figures 1 and 2: These look like a time series, not trends. There are no trend lines or values in these figures.

This has been corrected "trend" has been replaced by "time series"

Line 203: Higher rate than what?

Biofuel emissions have increased at a higher rate due than the other sources considered in the paper. This has been added in the text.

Line 211: "globally in Africa" is confusing. Which is it? Global or in Africa?

This sentence has been rewritten in lines 245-246 as:

"In Africa, all pollutant emissions are generally increasing …"

Line 227: What does "(4 over 7)" and "(1 over 5)" mean? Should "over" be "of"?

We replaced "over" by "of".

Lines 233-234: What are the numbers in brackets? Population or emission growth rates? Are the units correct? Should these be "% a-1"?

The numbers in brackets correspond to annual population growth rates. This sentence has been rewritten in lines 269-272:

"As for BC, the highest rates of increase in $SO_2$ emissions during this period are observed in regions where the population growth rate is the highest, i.e. West Africa (2.66% $yr^{-1}$), Middle Africa (3.10% $yr^{-1}$) and East Africa (2.71% $yr^{-1}$) whereas the lowest rates are found where the population growth rate is the lowest i.e. Southern Africa (1.64% $yr^{-1}$) and North Africa (1.87% $yr^{-1}$)."

Lines 240-241: Has the relative contribution of the different sectors changed over time?

The relative contribution of the different sectors shows small variations over time.

Line 255: What's the justification for looking at these countries specifically?

We found relevant to focus on countries in two regions that have different types of predominant sources (West African region and Southern African region), in order to discuss their specificities. We added "with different predominant sources" to the text to indicate the reason why these countries have been chosen in line 294.

Line 259: Is there a quantitative way to express the predominance of these two-wheel vehicles?

A new sentence was added in the text, which indicates the percentage of two-wheeled vehicles in the considered countries in lines 299-300:

"In Benin, TW vehicles represent 34% of road traffic emissions whereas of 16% only in Cote d'Ivoire for example."

Line 265: Is it industry or coal-fired power plants in South Africa that make the largest contribution? How does your estimate compare to emissions from Pretorius et al. (2015)?

Coal-fired power plants in South Africa correspond to the largest contribution of $SO_2$. This information is given in lines 269- 270 and compared with data of Pretorius et al. (2015). The following sentence has been added in lines 305-309:

"South Africa is the country emitting the highest amount of $SO_2$ with roughly 62% of the total African $SO_2$ emissions. Such an important contribution is due to coal-fired power plants with emissions in the range 1000-1500 Gg for the period 1999-2012. These numbers are in

agreement with the range given by Pretorius et al. (2015), i.e. 1500 – 2000 Gg for the same period."

Line 275: The Liousse et al. (2014) inventory projected dramatic increases in emissions in Africa. How do the % per year increase values in that inventory compare to your inventory?

The inventory of Liousse et al. (2014) based on 2005 emissions makes projections according to 3 scenarios until 2030. Our inventory available from 1990 to 2015 makes projections until 2030 following the emission trend obtained from 1990 to 2015, which are compared to those of Liousse et al. (2014). This is discussed in the last paragraph.

Line 278: Is $SO_2$ "slightly lower"? The difference is ~1.5 Tg. What accounts for this difference?

This sentence has been reformulated in lines 322-325:

"Fossil fuel and biofuel emissions from the DACCIWA inventory are slightly lower than those given by Liousse et al., (2014) for BC (0.64 instead of 0.69 Tg), $NO_x$ (5.08 instead of 5.80 Tg) and NMVOCs (8.33 instead of 8.60 Tg), lower for $SO_2$ (2.53 instead of 4.02 Tg) and slightly higher for OC (4.96 instead of 3.95 Tg) and CO (64.43 instead of 58.6 Tg). These differences may be explained by the use of more recent fuel consumption data and new emissions factors which are taken from direct measurements."

lines 280-281: The reasons for the difference are quite generic (could be said for any emission inventory comparison). Can you be more specific about the source and sizes of the differences in activity factors and emission factors?

The source and sizes of differences between activity data are very different from one sector to another and also from one country to another. This is the same for emissions factors which are very different from one pollutant to another, from one sector to another and also from one country to another. In this paper, we tried to explain the main differences observed between the emission inventories by giving some information on activity data and EF differences: all these data are not generally provided in the papers describing the inventories. A detailed comparison is planned in the coming years by the international GEIA (Global Emissions IniAtive) project, where the participants will give access to all their ancillary data.

Lines 285-286: Are these global emission inventories independent of each other for Africa?

It depends on the inventories: for example, CEDS$_{GBD-MAPS}$ and CEDS have a lot in common, and uses some data from the EDGAR datasets. There are also common data used in EDGAR and ECLIPSE.

Line 291: What accounts for these differences in BC and NOx (and likely other compounds too)?

As stated earlier, these differences arise from differences in the used emission factors and activity data : this was mentioned in lines 298 – 302.

Lines 299-302: Where do the other global inventories get their EFs from and by how much do these differ from the EFs used to develop your inventory?

EFs used in most global inventories are not publicly available, making the comparison difficult. Often, EFs in global inventories are taken from measurements performed in developed countries, which are not relevant for Africa.

Line 305: Higher by how much?

For example, we use 9.3 g/kg as the OC EF for wood for the residential sector, while Liousse et al. (2014) used 2.7 g/kg i.e. by a factor larger than 3.3. This is explained in Keita et al., (2018).

Line 320: Using the REF and CCC* acronyms immediately without first defining them assumes the reader knows what these are.

The names of the scenarios have been explained. The new sentences, in lines 381-386 are:

"Finally, Liousse et al. (2014) have estimated the emissions of BC and OC in 2005 and 2030, using different scenarios. The authors defined the REF scenario as the state of the world for "business and technical change as usual" conditions, driven solely by basic economics. Another scenario has been proposed (called CCC*), where the introduction of carbon penalties and African specific regulations are implemented to achieve a large reduction in emissions from incomplete combustion. These values are shown by the dots in Figure S2 for the REF and CCC* scenarios.  We have linearly extrapolated the DACCIWA emissions for these two species, as shown by the plain lines in Figure S2."

Section 3.4: Are subsections (3.4.1-3.4.2) is this section necessary? Each subsection is only a paragraph long.

We have chosen two sub-sections to clearly show the description of the methodology and the uncertainties on the results.

Line 358: Would emissions always be overestimated? Is this because combustion efficiency varies with development stage? Wouldn't this lead to an overestimate for some emitted compounds and underestimate for others?

The combustion efficiency varies with technology, depending on the stage of development of each country. Consequently, the choice of development level to determine EF values could lead to an overestimation for some emitted compounds and an underestimation for others. However, there is a lack of ancillary data in Africa, to correctly handle these details.

Figure 3: "-1" in "cell-1" should be superscripted.

This has been corrected.

Figure 6: Consider including a line starting from 2005 for the projected emissions from the Liousse et al. inventory.

The projected emissions from the Liousse et al. (2014) inventory and this inventory are shown in figure S2 in the supplementary material.

Figure 7: What is the second "**D**" in "EDGAR**D**4.3"

This has been corrected.

Table 1: Is this table necessary, when the only difference is the daily consumption? This could be more succinctly stated in the text.

This table has been deleted and stated in the text as suggested

Table 2: Should "CH" be "charcoal use", rather than just "charcoal"?

"CH" was the fuel "charcoal" rather than "charcoal use".

Table 3: Fix "Gg. year-1".

This has been done

---

## Author Comment (AC2) · 19 Apr 2021

**Review of "African Anthropogenic Emissions Inventory for gases and particles from 1990 to 2015" by S. Keita et al. submitted to ESSD in 2020**

By Keita et al.

**Response to Reviewer's comments**

Dear Editor,

First, we would like to thank the reviewers for their positive comments and their suggestions to improve the quality of this document. All the questions were treated and our document was fully revised taking into account all the reviewers' comments. The paper, figures and tables were modified as a result of the different suggestions and remarks.

We have indicated the referees' comments in black and the authors' responses in blue.

**Reviewer's comments**

**Referee #2**

**Anonymous Referee #2**

In this paper the authors provide a very valuable emission inventory for a region for which very little information is available with regard to anthropogenic emissions. This is a well-written and original paper that I must be accepted for publication in a journal such as ESSD. There are, however, a number of typographical errors throughout the paper, which must be addressed by the authors. This can be rectified through subjecting the paper for language editing by a English language editor, as well as assistance with text editing by a native English speaker with a background in atmospheric science.

We thank Referee #2 for providing very useful comments and suggestions on the manuscript. Following the referee' comments, typographical errors have been corrected throughout the document and proofreading has been carried out with the help of a native English speaker with a background in atmospheric science as suggested by the referee. Many changes were made in the manuscript, and have been taken into account following the referee's comments.

---

## Author Response (AR2)

**Review of "African Anthropogenic Emissions Inventory for gases and particles from 1990 to 2015" by S. Keita et al. submitted to ESSD in 2020**

By Keita et al.

**Response to Editor comments**

Dear Editor,

First, we would like to thank the Editor and reviewers for their positive comments and their suggestions to improve the quality of this document. All the questions were treated and our document was fully revised taking into account all the comments. The paper and figures were modified as a result of the different suggestions and remarks.

We have indicated the Editor comments in black, the author responses in blue and the changes in the revised manuscript in red.

**Editor comments**

Topical Editor Decision: Publish subject to minor revisions (review by editor) (18 May 2021) by Mauricio Osses

Comments to the Author:

Please correct the format of series in Figures 4, 5 and 7.

1) Make sure the name of each sector is always the same. Figure 5 says "Other sectors" while other figures are named "Other sector".

Thanks for this observation, the name is "other sectors" : this has been corrected in line 303, Figures 4, 5 and 7.

2) Waste and Flaring are indicated with/without capital first letter, harmonise all names with the same format.

This has been corrected in Figure 4, 5 and 7

3) I suggest using the same series color for the same sectors. Figures 4 and 5 are different than figures 7a/7b, but the sectors are still the same. Ideally, use the same series order to facilitate comparison.

The same series color are now used for the same sectors for Figures 4a/4b, 5 and Figures 7a/7b.